# Evaluating the Heterogeneous Impacts of Adoption of Climate-Smart Agricultural Technologies on Rural Households' Welfare in Mali

Bola Amoke Awotide [1,*], Adebayo Ogunniyi [2], Kehinde Oluseyi Olagunju [3], Lateef Olalekan Bello [4], Amadou Youssouf Coulibaly [5], Alexander Nimo Wiredu [6], Bourémo Kone [7], Aly Ahamadou [7], Victor Manyong [8] and Tahirou Abdoulaye [9]

1    Centre for Agrarian Transformation and Development (CATD), Bamako 91094, Mali
2    International Fund for Agricultural Development (IFAD), Abuja 90021, Nigeria
3    Economics and Evaluation Branch, Department of Agriculture, Environment and Rural Affairs, Belfast BT4 3SB, UK
4    Department of Global Agricultural Science, The University of Tokyo, Tokyo 113-8657, Japan
5    Institut de Pedagogie Universitaire (IPU), Bamako 91094, Mali
6    AKDE Solutions Ghana Ltd., Accra P.O. Box LG 68, Ghana
7    Institut d'Economie Rurale (IER), Bamako 91092, Mali
8    Social Science and Agribusiness, International Institute of Tropical Agriculture (IITA), Dar es Salam 34441, Tanzania
9    Social Science and Agribusiness, International Institute of Tropical Agriculture (IITA), Bamako 91094, Mali
*    Correspondence: catdrng@gmail.com or bolatide2014@gmail.com

**Abstract:** Climate change is negatively affecting agricultural production in the Sahel region. Climate-Smart Agricultural Technologies (CSATs) are disseminated to reduce these negative effects, and particularly those on resource-poor farm households. This article investigates the distributional impacts of the adoption of CSAT on-farm households' welfare using a dataset that covers four regions, 32 communes, 320 villages, and 2240 households in Mali. Using an instrumental variable quantile treatment effects model, the paper addresses the potential endogeneity arising from the selection bias and the heterogeneity of the effect across the quantiles of the outcome variables' distribution. The results show that the adoption of CSAT is positively associated with improved households' welfare. The farmers' decision to adopt any CSAT is influenced by access to credit, contact with extension agents, participation in training, access to information through the television, and being a member of any organization such as a cooperative society. Moreover, the effect of the adoption of CSAT on household welfare varies across the different households. In particular, the results show that the impact of the adoption of CSAT on households' welfare is generally higher for the poorest (farmers located at the bottom tail of the distribution) end of the welfare distribution. The findings, therefore, highlight the pro-poor impact of the adoption of CSAT in the rural Malian context, as well as the need to tailor the CSAT interventions toward specific socio-economic segments of the rural population in Mali.

**Keywords:** Climate-Smart Agricultural Technologies; quantile regression; endogeneity; Sahel region; Mali

## 1. Introduction

The global population is rapidly growing and is projected to reach 8 billion people in 2022 [1]. It is also predicted to reach 9.2 billion by 2050 [2], with a projected increase in food demand of 59–102% [3,4]. Efforts to increase agricultural productivity by 60–70% are necessary to provide adequate food for the population in 2050 [2]. Agricultural productivity growth is one of the most critical and effective pathways for agricultural research and technologies to increase rural incomes and reduce poverty [5]. Studies have shown that

agricultural growth has a more considerable effect on reducing poverty than growth in any other sector [6–8]. However, agricultural productivity in Africa is among the lowest in the world. More importantly, sub-Saharan Africa (SSA) remains the world's most food-insecure region, with almost one-fourth of the inhabitants—over 230 million—being undernourished [9]. Although agriculture occupies a vital position in the economies of most SSA regions, employing more than 75% of the population, it is mainly rainfed [10–13], and hence, highly susceptible to climate change effects [14]. Most of the SSA population depending on agriculture as a source of livelihood comprises smallholder farmers who are the most highly vulnerable to the negative effects of climate change [15,16], such as crop failure, yield loss, and soil deterioration [17]. This suggests that climate change may have a greater impact on underdeveloped nations, rural economies, and agricultural production. Failure to act and mitigate the effect of climate change can worsen the already appalling poverty and food insecurity situations of most of the rural households whose survival depends solely on agriculture [18].

Future projections based on observed climate trends indicate that temperatures in SSA are consistently rising at an alarming rate compared to the global average increase during the 21st century [19–22]. Therefore, it is most likely that SSA might be strongly affected by climate change. Furthermore, Africa's agrarian economies are likely to disproportionately bear the burden of substantial agricultural yield losses [23,24]. Thus, failure to act and mitigate the effect of climate change can worsen the already appalling poverty and food insecurity situations of most of the rural households whose survival depends solely on agriculture [18], and further undermine their welfare.

The present situation in the Sahelian regions of Africa in the face of erratic climate change effects is highly problematic. The incidence of drought and floods in this region is becoming more severe and frequent over time. Several factors have been identified to be responsible for the vulnerability to climate change in the Sahel. Notable among these factors is the pervasiveness of poverty, which reduces the resources with which the affected communities, households, and individuals can adapt to and recover from climate events. Other factors are the over-reliance of most of the population (80–90%) [25] on farming and pastoralism, linked intimately to weather trends and environmental conditions. This implies that climate change effects can jeopardize the livelihood and food security of the Sahelian rural households.

The Republic of Mali, a landlocked country in the Sahelian region of West Africa, is highly negatively affected by climate change, leading to agricultural production reduction. Increasing agricultural productivity through the adoption and diffusion of modern agricultural technologies is recognized as one of the key pathways for economic and agricultural transformation in developing countries [26,27], particularly in regions experiencing huge adverse effects of climate change. Hence, there is a great need to concentrate on technologies such as Climate-Smart Agricultural Technologies (CSATs) capable of mitigating the adverse effects of climate change, The Climate-Smart Agriculture (CSA) concept was proposed by the Food and Agriculture Organization of the United Nations (FAO) at the Hague Conference on Agriculture, Food Security, and Climate Change in 2010. CSA is an approach to agricultural development that aims to address the intertwined challenges of food security and climate change [28]. It is built upon three pillars: increase agricultural productivity and incomes, adapt and build resilience to climate change within the agricultural systems, and reduce greenhouse gas (GHG) emissions when possible. CSAT programs' promotion of different technologies, practices, and policies involves diverse institutions and investments. These interventions take place at the field, farm, regional, and national levels [29]. Therefore, the adoption of CSAT is viewed as a way of combatting the adverse effects of climate change on agricultural productivity.

In trying to better understand the adoption of CSAT, [30] used the Average Treatment Effect to derive the consistent parametric estimators of the potential adoption rates of CSAT and practices in a Climate-Smart Village (CSV) site of Mali. The study revealed that the observed adoption rate varied from 39 to 77% according to the CSAT options, while the po-

tential adoption rates of the technologies and practices ranged from 55 to 81%. This implies an adoption gap of 2% to 16% attributed to incomplete diffusion of CSAT practices, but provides no information about the impact of CSAT adoption. A large piece of evidence from the literature shows that adopting such technologies in developing economies contributes to increasing farm productivity and reduces household poverty [31–33]. Concerning CSAT, an extensive literature exists on the adoption impacts of individual climate-smart practices, although with divergent findings. For instance, Di Falco and Chavas [34] reported that biodiversity positively affects risk reduction among barley producers in Ethiopia. In a related study, Di Falco and Veronesi [35] reveal that adaptation to climate change, through the adoption of soil conservation, changing crop varieties, switching from early to late planting, and other practices, generates an increase in maize yield among the adopters' farm households in Ethiopia. Other studies have shown that soil conservation, crop choice, and other practices can increase technical efficiencies among farmers and minimize on-farm environmental damage [36–38]. This implies that the results from these studies are not only mixed, but that they are also inconclusive. Most importantly, many of these studies only investigate the effect of the adoption of CSAT on mean yield, mean income, and expenditure. This general result only implies that the adoption of CSAT tends to have a statistically significant positive impact on the income and total household expenditure of the "average" (or the mean income/expenditure) farmer. This finding does not give specific information on whether (and how much) the adoption of CSAT affects the income and consumption expenditure at the lower or upper end of the distribution. This suggests a gap exists in the literature about the potential heterogeneity impact of the adoption of CSAT. More importantly, the overarching question is whether the adoption of CSAT has a heterogeneous impact on households at different points of the income and total household expenditure distribution. Of great importance in many areas of empirical economic research is the ability to understand or provide answers to the effect of any intervention on the distribution of outcomes.

Overall, policymakers in developing countries would be more interested in supporting increased adoption of CSAT if there is empirical evidence that the lower-welfare farmers (who are typically poor smallholders) specifically benefit from CSAT adoption. Understanding the effects of CSAT adoption at different points of the welfare distribution would provide a more detailed insight into the economic impacts of CSAT adoption. Suppose the adoption of CSAT has a statistically significant effect on the higher end of the welfare distribution but reveals no negative impact on the lower welfare/poor smallholder farmers. This would imply that investing in the development, dissemination, and promotion of the CSAT is not a good policy option, especially in cases where the priorities are to increase farm productivity and the welfare of smallholder households. If, on the other hand, the findings show that the adoption of CSAT significantly impacts the productivity, income, and welfare of the farm households in the lower tail of the distribution, there will be adequate justification to promoting the adoption of CSAT among the resource-poor smallholder farmers. In addition, it might further imply that it could increase the poor smallholders' income, increase agricultural productivity, and improve the overall farm household welfare.

The primary motivation of this study is the estimation of the overall impact of the adoption of CSAT on income and household welfare, and the heterogeneous effects on the farm household's welfare. The study seeks to fill these gaps by addressing the following research questions: What determines households' decisions to participate in the adoption of CSAT? What is the overall collective impact of the adoption of CSAT on income and household welfare? Findings from previous studies suggest that observably both lower-yielding and higher-yielding farmers in developing countries equally benefit from the adoption of CSAT, regardless of the fact that developing-country farmers at the lower end of the yield distribution tend to be poorer than those at the upper end. These studies suggest that the benefits of CSAT adoption would be felt equally by all types of farmers. Specifically, we hypothesize that the impacts of CSAT adoption differ across the distributions of our

welfare indicators, including income, per capita total expenditure, and food and non-food expenditure. One way to capture the effects of CSAT adoption at different points of the welfare distribution is to use the quantile regression technique introduced by Koenker and Bassett [39]. This technique has been used in various studies in applied economics to examine the effects of policy instruments at different points of a particular outcome distribution, mainly in studying wage distribution or trade effects; see [40–42]. However, if there are endogeneity or self-selection problems, the coefficient estimates from standard quantile regression techniques may be biased [43–45]. Moreover, the standard instrumental variable (IV) or two-stage least squares (2SLS) approach in ordinary least squares (OLS) regression is not directly applicable in a quantile regression context. Chen and Portnoy [46] developed a quantile regression analog to the standard 2SLS approach called a two-stage quantile regression (2SQR). However, Chernozhukov and Hansen [45] show that 2SQR is inconsistent when the quantile treatment effect differs across quantiles.

Chernozhukov and Hansen [45,47,48] developed an IV technique applicable for quantile regressions (called the instrumental variable quantile regression or IVQR) to address this problem. They have shown that the estimated coefficients in this approach are unbiased, e.g., [44,49,50]. To the best of our knowledge, the estimation of the possible heterogeneous effects of CSAT adoption on different points of the welfare distribution, especially in the presence of self-selection (i.e., non-random selection of CSAT adopters) is still not a well-researched area, particularly for the Sahelian region of West Africa. Contributing to the literature by filling this gap is one of the primary motivations for this study. Specifically, we seek to determine the effect of CSAT adoption at different points of welfare distribution. To achieve this objective, we adopted the instrumental variable quantile regression (IVQR) of Chernozhukov and Hansen [47] and Chernozhukov and Hansen [51] to identify the quantile treatment effect. Explicitly, the instrumental variable quantile regression model [47,52] aims to investigate heterogeneous treatment effects in the presence of an endogenous binary treatment variable.

The rest of this paper is outlined as follows: Section 2 presents the materials and methods. Section 3 presents the results and discussion. The conclusion of the study is presented in Section 4.

## 2. Materials and Methods

### 2.1. Data Collection and Sampling Framework

To investigate the economic impacts of the adoption of CSAT, we used primary survey data collected from the main crop-producing regions of Mali between June and October 2019. We conducted the survey on a total of 2240 farm households in different villages selected from 32 communes that cut across 4 regions of Mali. A multi-stage sampling procedure was used for the selection of the targeted sample. First, four regions were purposively selected (based mainly on accessibility, security, and willingness of the community to participate in the project) from the regions in Mali. Eight communes were selected from each of the regions, making a total of 32 communes. From each of these communes we randomly selected 10 villages per commune, and then seven farm households were selected from each of the villages. In the communes, most of the systemic issues (local policies, culture, demographic, socio-economic, and agroecology) are similar, and all the selected households have agriculture as the main occupation, producing crops, and some also rearing animals in addition to crop production.

A structured questionnaire was prepared and carefully administered to gather household-level primary data. Well-trained enumerators collected the data in face-to-face interviews. The data were collected on households' demographic characteristics, sources of livelihoods, food security status, off-farm employment, asset ownership, types and quantities of crops produced, sale of crops and output prices, household access to credit, markets, and extension services, and membership of producers' associations, among many others. Moreover, the data included information on types and volume of inputs used in crop production, input supply arrangements, costs of inputs (hired labor, fertilizers, pesticides,

and improved seeds), improvement practices, market outlets, and overall production and marketing challenges.

## 2.2. Conceptual Framework and Estimation Strategy

The current and future impacts of climate change are major sources of concern in Sub-Saharan Africa (SSA), due to the predominance of rain-fed subsistence agriculture in the region [11,12]. The region is affected by extreme weather events and by long-run climate variability, which can severely reduce yields and increase the levels of uncertainty with respect to agricultural production and output prices. The situation can further lead to an overall increase in the welfare vulnerability of smallholders [53], hence the need for Climate-Smart Agricultural Technology (CSAT). The broad definition of CSAT includes the integration of different farming/agronomic practices and systems, as well as the improvement of input use, such as seeds, pesticides, and water. It includes typical technologies such as climate-stress-tolerant seed, irrigation, and fertilizer, which are classic examples in technology adoption studies [54,55], as well as practices such as intercropping, conservation agriculture, manuring, and water harvesting; elsewhere, these are discussed under terms such as sustainable practices or conservation agriculture [56,57]. Essentially, CSAT and practices contribute to the adaptation of farmers to the effects of climate change and, more importantly, it helps resource-poor farmers to address climate change issues such as extreme drought, extreme precipitation, and changes in seasonal timing. In this regard, the ultimate aim of CSAT is to simultaneously increase agricultural productivity and resilience in the face of climate change, while at the same time reducing greenhouse emissions from agricultural systems [28].

Evidence from the literature shows that the adoption of locally adapted CSAT portfolios can lead to an increase in productivity of between 7 and 18% [54,58]. Additionally, CSAT options typically reduce production risk by increasing the resilience of the agricultural system [28]. As Teklewold et al. [59] show for Ethiopia, and Arslan et al. [60] for Zambia, adoption rates of CSATs among smallholders often remain low, despite the potential of CSAT to increase productivity and resilience [61]. The decision of the rural farm households to adopt CSAT and practices is modeled under the assumption that most farmers are rational and risk-averse, and therefore will always act to maximize expected profit. According to Feder et al. [62], farm households adopt new technology when they expect a more profitable outcome than what they gained from the existing traditional technologies or other previously available technologies. Therefore, CSAT will only be appealing to households experiencing climate change effects if the expected benefits significantly compensate for the costs. Hence, households' decision to adopt CSAT may be viewed through the lens of constrained optimization where the household chooses the technology if it is available, and affordable, and its usage is expected to be beneficial [63].

First, we specified the drivers of farm households' decisions to adopt CSAT. Many studies, e.g., [64–67], have assessed the factors that influence farm households' decisions to adopt any new improved agricultural technology with either probit or logit models. The two models are based on the normal and logistic cumulative distribution functions, respectively. Although the models are similar, the logistic distribution has slightly fatter tails. In this study, we fit the binary probit model to estimate the farm households' decisions to adopt CSAT since the response-dependent variable (adoption of CSAT) is binary. The probit model can resolve the problem of heteroscedasticity and satisfies the assumption of a cumulative normal probability distribution [68]. In addition, the probit model also includes believable error term distribution as well as realistic probabilities [69].

The independent variables included in the model are age, education group membership, farm size, etc. Therefore, the probit model is specified as shown below:

$$G_i = F(M_i\gamma) + \mu_i \tag{1}$$

$$G_i = \begin{cases} 1, & if\ adopted\ CSAT \\ 0, & otherwise \end{cases}$$

where $\mu \sim N(0,1)$; $\gamma$ = maximum likelihood; $\mu_i$ = error term; $M$ = set of independent variables included in the model. In the case of normal distribution function, the model to estimate the probability of observing a farmer using CSAT can be stated as:

$$P(G_i = 1|M) = \phi(M'\gamma) = \int_{-\infty}^{m'\gamma} \frac{1}{\sqrt{2\pi}} exp\left(-z^2/2\right) dz \qquad (2)$$

where $P$ is the probability that the $i$th farm household will adopt any of the disseminated CSAT, and 0 otherwise. Since the estimates of the probit model provide only the direction of effects, the marginal effects are usually calculated to interpret the actual change in the probability of independent variables:

$$\text{Marginal effects} = \gamma_i \, \phi(z) \qquad (3)$$

where $\gamma_i$ = coefficient of the variables; $\phi(z)$ = the cumulative normal distribution value associated with the mean dependent variable from the probit estimation. Evaluation of the impact of adoption of CSAT on the distribution of welfare outcomes required the estimation of the conditional linear quantile model as follows:

$$J_i^{\pi} = M_i \alpha^{\pi} + G_i \beta^{\pi} + \varphi_i \qquad (4)$$

where $\beta^{\pi}$ denotes the quantile treatment effect (QTE) of adoption of CSAT ($G_i$). $J_i^{\pi}$ corresponds to the $\pi$th quantile of the distribution of the welfare outcomes. $M_i$ is a vector of observed covariates including socio-economic/demographic characteristics, etc.; $\alpha^{\pi}$ is a vector of parameters of the covariates to be estimated; $\varphi_i$ is the unobserved random variable. However, the treatment (adoption of CSAT) is non-random in the population, implying that adoption of CSAT may be potentially endogenous to the outcome variables; thus, using Equation (4) may lead to erroneous impact estimate.

Following Olagunju et al. [50], Chernozhukov and Hansen [51], Okumu and Muchapondwa [70], and Abadie et al. [71] we examine the impact of adoption of CSAT on the distribution of welfare outcomes, measured in terms of per capita total households' income, and per capita total expenditure (food and non-food) employing the QTE conditional on covariates, as originally developed by Abadie et al. [71] We specify the empirical econometric model of the Abadie et al. [71] conditional IV-QTE model as follows:

$$\beta\left(\hat{\pi}_{IV}, \hat{\delta}\pi_{IV}\right) = argmin \sum W_i^{AAI} \cdot \rho_{\pi}(T_i - M\beta_i - G_i\delta) \qquad (5)$$

where: $W_i^{AAI} = 1 - \frac{G_i(1-L_i)}{1-pr(L=1|M_i)} - \frac{(1-G_i)L_I}{pr(L=1|M_i)}$.

Determining the QTE in Equation (5) requires the use of an instrumental variable (IV) to obtain a consistent estimate of the treatment effect. However, the main concerns with respect to an IV are weak instruments and over-identification. Moreover, if the instrument affects the farm households in various ways (heterogeneity), the resultant treatment effects may be problematic [72]. In this study, a valid IV must be strongly correlated with the farmers' decision to adopt CSAT and being uncorrelated with the outcome variables is highly required. Past studies on adoption and its impact on various outcomes are of the opinion that no farm household can make any new technology adoption decision without first having adequate information about the technology. Being aware of a new technology has been advocated as a valid IV for the estimation of the adoption impact of new technology.

Intuitively, the farmers' awareness about the capability of CSAT to mitigate the negative effects of climate change, particularly in relation to drought tolerance, and early/extra-early maturing improved varieties in places where erratic rainfall, drought, and flood are big challenges can positively influence the farm households' decision to adopt CSAT. However, being aware of the existence of a new technology and its potential to increase productivity cannot impact the farm households' welfare. The farm households' welfare

can be impacted only if the farm households made an active decision to adopt CSAT. Thus, awareness of CSAT fulfilled the exclusive restriction for it to be a valid IV in this study, where $L$ is the IV (awareness of CSAT).

Equation (5) is estimated using the IV-QTE command in STATA because it produces analytical standard errors that are consistent even in the case of heteroscedasticity [72]. Given that some weights may be negative or positive, the *ivqte* stata command uses the local logit estimator and implements the Abadie, Angrist, and Imbens (AAI) estimator with positive weights. A substitute offered by Abadie et al. [71] demonstrates that the following weights can be implemented as another option to $W_i^{AAI}$, where $W_i^{AAI} = E\left[W_i^{AAI}|G_i, T_i, M_i\right]$, and are always positive. The IV-QTE utilizes the local linear regression to estimate $W_i^{AAI}$.

*2.3. Variables*

The Treatment and Welfare Outcome Variables

The treatment variable in this study is adoption of CSAT and practices, and it is defined as 1 if the farm household adopts any of the Climate-Smart Agricultural Technologies (early/extra early improved seed varieties, irrigation, etc.) or practices (intercropping, zero tillage, soil and water management, integrated pest-management practices, etc.), and zero otherwise. The outcome variable is welfare, which we proxied with income, per capita total expenditure, and food and non-food expenditure.

## 3. Results and Discussion

*3.1. Variable Definition and Descriptive Statistics*

Table 1 presents the descriptive statistics of selected variables for the empirical analyses showing that a considerable percentage of the sampled households (97%) have farming as their primary occupation, indicating the importance of agriculture in the livelihood portfolio of the study area. About 91% of the sampled households are aware of improved agricultural technologies. About 93% of the farm households reported that they had received information about these agricultural technologies through formal sources of information that comprise radio, television, newspaper, contact with extension agents, and participation in different trainings organized by research institutes and NGOs. This shows that, over the years, the various extension outlets have been effective at creating awareness about improved agricultural technologies and practices. This could have encouraged the majority of the rural households—about 75%—to adopt at least one improved agricultural technology. However, the adoption of CSAT is about 61%. In terms of demographic characteristics, about 99% of the sampled households are male-headed. The household head's average age is 56 years and, therefore, they are considerably experienced in agriculture. The average household size is seven persons.

Rural farm households' opportunities to participate in development programs and access to land for agricultural production in most cases depend on the households' residence status in the selected project intervention villages. Almost all the sampled households (98%) are 'natives', i.e., residing in their respective villages for an average of 55 years. In addition, a significant percentage of the farm households (89%) own land for farming, and the estimated average total land area available for farming is 13.51 ha, out of which only 8.31 ha (61.5% of the total land area) is currently under staple crops production. The result further reveals an average land pressure of two persons per hectare. This indicates that the farmers could be having some challenges related to land access for large-scale farming, and it is a pointer to the need for the farm households to adopt improved agricultural technologies to move away from extensive, to intensive, agricultural production. Only about 39% of the household heads are literate, with an average of about six years of schooling. About 81% of the households are members of an organization. Although the majority of the household heads do not have formal education, available institutions such as extension and famer associations enabled the households to access useful information about agriculture.

**Table 1.** Variable definitions and descriptive statistics.

| Variable | Description | Mean (Std. Dev.) |
|---|---|---|
| The main occupation of the household head | 1 if the main occupation of the household head is farming, 0 otherwise | 0.97 (0.18) |
| Adoption of CSAT | 1 if the farmer adopts any CSAT technology, 0 otherwise | 0.61 (0.49) |
| Per capita consumption expenditure | Per capita consumption expenditure (CFA) | 107,739.8 (105,209.8) |
| Gender | 1 if the farmer is male, 0 otherwise | 0.99 (0.09) |
| Age | Age of the household head in years | 56.39 (14.77) |
| Residence status | 1 if the farmer is a native of the village, 0 otherwise | 0.98 (0.15) |
| Household size | Number of family members | 7.57 (5.74) |
| Education | Number of years of formal education | 6.39 (4.35) |
| Owned land | 1 if the farmer-owned land, 0 otherwise | 0.89 (0.30) |
| Total land area | The total land area available for crop production (hectares) | 13.51 (10.56) |
| Average cultivated farm size | The average farm size is currently under crop production (hectares) | 8.31 (5.84) |
| Access to extension | 1 if the farmer has access to extension, 0 otherwise | 0.73 (0.44) |
| Access to credit | 1 if the farmer has access to credit, 0 otherwise | 0.33 (0.47) |
| Own a bank account | 1 if the farmer owns a bank account, 0 otherwise | 0.1381 (0.345) |
| Main income source | 1 if the main income source is agriculture, 0 otherwise | 0.609 (0.488) |
| Distance to the nearest market | Distance of farmer to nearest market (km) | 16.33 (24.92) |
| Distance to the nearest village | Distance of farmer to the nearest village (Minutes) | 25.57 (46.01) |
| Residence in the village | Number of years of residence in the village | 55.21 (21.28) |
| Farming experiences | Number of years of farming experience | 37.88 (17.42) |
| Literacy rate | 1 if the farmer can read or write in French | 0.39 (0.49) |
| Awareness of improved agricultural technologies | 1 if the farmer is aware of any of the improved technologies, 0 otherwise | 0.91 (0.29) |
| Awareness of CSAT technologies | 1 if the farmer is aware of CSAT technologies and practices | 0.80 (0.40) |
| Formal sources of information | 1 if the farmer receives information from formal sources, 0 otherwise | 0.93 (0.26) |
| Membership of organization | 1 if the farmer is a member of any organization, 0 otherwise | 0.81 (0.39) |
| Migrant household | 1 if at least one person has migrated from the household, 0 otherwise | 0.49 (0.50) |
| Attended training | 1 if the farmer has participated in any training, 0 otherwise | 0.24 (0.43) |

### 3.2. Test of Mean Differences in Welfare Outcome

In this section, we carried out an observed evaluation of the indicators to uncover the difference in all the selected welfare indicators between the adopters and non-adopters of the CSAT, and test if the differences are statistically significant. The results as presented in Table 2 show that the farm households that adopted the CSAT are not similar to the non-adopters of CSAT in many of the indicators. The farm households that adopted CSAT technologies appear to have significantly higher values of all the selected indicators, except for the non-farm income, where the two groups of households seem to be similar.

The simple comparison of the means of these selected welfare indicators for the adopters and non-adopters does not imply an impact of the adoption of CSAT technologies on the households' welfare. This is because the presented observed differences might be due to other observed and unobserved factors that have nothing to do with the adoption of CSAT. In other words, the observed difference in the mean outcomes between the two groups (adopters and non-adopters) can be attributed to both the impact of adopting the improved agricultural technologies and pre-existing differences (selection bias) [73]. Thus, the observed differences in all the outcomes between the adopters and non-adopters have no causal interpretation. Consequently, to empirically determine the impact of adopting the CSAT technologies on welfare, we adopted the IV-QTE.



**Table 2.** Test of mean differences in welfare indicators, by adoption status.

| Variable | Total N = 2186 | CSAT-Adopters N = 1332 | CSAT Non-Adopters N = 854 | Mean Difference | *t*-Test |
|---|---|---|---|---|---|
| Total household income (CFA) | 412,929.50 (9244.65) | 452,091.70 (12,297.35) | 351,847.40 (13,607) | 100,244.40 (18,830.41) | 5.32 *** |
| Per capita total household income (CFA) | 70,389.53 (1695.37) | 77,446.92 (2242.526) | 59,549.31 (2529.79) | 17,897.62 (3448.36) | 5.19 *** |
| Total income from crop production (CFA) | 247,195.2 (8752.77) | 288,255.80 (12,100.72) | 183,152.30 (11,750.23) | 105,103.50 (17,802.29) | 5.90 *** |
| Total non-farm income (CFA) | 131,292.40 (5172.13) | 131,363.40 (6774.14) | 1,311,081.60 (7982.47) | 181.74 (10,603.23) | 0.02 |
| Total consumption expenditure (CFA) | 599,729.60 (11,231.95) | 645,263.00 (14,527.07) | 528,710.20 (17,432.09) | 116,552.90 (22,890.81) | 5.09 *** |
| Per capita consumption expenditure (CFA) | 108,136.80 (2275.65) | 117,921.50 (2972.19) | 93,107.50 (3469.51) | 24,813.98 (4626.67) | 5.36 *** |
| Total non-food expenditure (CFA) | 658,591.20 (24,009.00) | 702,929.40 (26,519.97) | 589,436.00 (45,372.88) | 113,493.40 (49,160.17) | 2.31 ** |
| Total food expenditure (CFA) | 48,441.57 (3139.45) | 59,579.30 (4782.06) | 31,069.84 (2896.42) | 28,509.46 (6407.10) | 4.45 *** |
| Total Farm size (ha) | 13.49 (0.24) | 13.64 (0.30) | 13.23 (0.39) | 0.41 (0.49) | 0.84 |
| Total monetary value of household asset value (CFA) | 164,2478.00 (33,621.01) | 1,691,304.00 (42,759.84) | 1,566,323.00 (54,319.81) | 124,980.80 (68,873.49) | 1.81 * |
| Total monetary value of productive assets (CFA) | 847,241.20 (15,664.93) | 919,047.80 (20,070.50) | 735,243.10 (24,586.09) | 183,804.70 (31,872.43) | 5.77 *** |

Note: The figures in parentheses are standard errors. *** $p < 0.01$, ** $p < 0.05$, * $p < 0.1$.

*3.3. Determinants of Adoption of Climate-Smart Agricultural Technologies*

In this section, we examine the factors influencing the farmers' decisions to adopt CSAT, as shown in Table 3. Overall, the results confirm that a farmer's adoption decision is influenced by socioeconomic and demographic characteristics (individual and household level), social capital, institutional support to the farmers, and farm-level susceptibility to climate change.

The results show that farmers' participation in climate change-related training has a significant and positive relationship with the adoption of CSAT. The marginal effect suggests that participation in training increases the likelihood of adoption by 14.7%. The probable reason for this, as noted in previous studies [74,75], is that training provides an exposure mechanism that allows farmers to have a clearer understanding of the processes and procedures of the technologies. We found that there is a negative association between farming experience and farmers' adoption decisions. Studies such as Ogunniyi et al. [76] and Sardar et al. [77] noted that the probability of adopting improved agricultural technologies decreases with increasing farmer experience and may be due to risk aversion.

The results also suggest that household size reduces the probability of adoption of CSAT. This is consistent with previous studies by Baiyegunhi et al. [78] and Zhang et al. [79] that found that households with large sizes are less likely to adopt climate-smart technologies. In the same vein, Mahama et al. [80] note that large households often face a challenge of intra-household budget allocation in which food expenditure takes a large share of total household allocation, leaving less for other farming expenditures such as improved inputs. The results show that the likelihood of adopting the CSAT decreases with the increasing size of farms. This may be attributed to the fact that farmers may consider the cost of adopting the technology on a large farm size without evaluating the economies of scale that can be beneficial due to the large expanse of land. Another reason is that they may consider their large production area enough for their needs, so have less motivation for adoption.

**Table 3.** Estimates of determinants of adoption of CSAT.

| Variables | Probit Regression | | Marginal Effects | |
|---|---|---|---|---|
| | Coefficient | Std. Error | dy/dx | Std. Error |
| Number of years of residence in the village | 0.003 | 0.002 | 0.001 | 0.001 |
| Attend training (yes = 1) | 0.404 *** | 0.082 | 0.147 *** | 0.028 |
| Years of farming experience | −0.013 *** | 0.002 | −0.005 *** | 0.001 |
| Tropical Livestock Unit (TLU) | 0.002 | 0.002 | 0.001 | 0.001 |
| Literacy (yes = 1) | 0.031 | 0.066 | 0.012 | 0.025 |
| Total farm size (ha) | −0.006 * | 0.003 | −0.002 * | 0.001 |
| Household size | −0.013 ** | 0.006 | −0.005 ** | 0.002 |
| Access to information (television = 1) | 0.216 *** | 0.065 | 0.082 *** | 0.024 |
| Access to credit (yes = 1) | 0.278 *** | 0.080 | 0.103 *** | 0.029 |
| Age of household head (years) | 0.000 | 0.013 | 0.000 | 0.005 |
| Square of age | 0.000 | 0.000 | 0.000 | 0.000 |
| Contact with extension agents (yes = 1) | 0.423 *** | 0.070 | 0.164 *** | 0.027 |
| Household with migrant (yes = 1) | −0.047 | 0.063 | −0.018 | 0.024 |
| The main income from agriculture (yes = 1) | 0.240 *** | 0.064 | 0.092 *** | 0.025 |
| Married (polygamous = 1) | −0.066 | 0.069 | −0.025 | 0.026 |
| Distance to the nearest village (km) | 0.001 * | 0.001 | 0.001 * | 0.000 |
| Walking distance to the nearest market (min) | −0.000 | 0.001 | −0.000 | 0.000 |
| Membership of any organization (yes = 1) | 0.306 *** | 0.081 | 0.119 *** | 0.032 |
| Bank account (yes = 1) | −0.234 ** | 0.102 | −0.091 ** | 0.040 |
| Constant | −0.241 | 0.372 | | |
| Number of observations | 2216 | | 2216 | |

Note: *** $p < 0.01$, ** $p < 0.05$, * $p < 0.1$.

The results of the institutional variables used in the model suggest that the adoption rate can be improved if farming households receive certain support from relevant agencies. For instance, we found that access to information on climate change (and its impact) via mass media was positive and significantly influenced the adoption of CSAT. The marginal effect shows that the probability of adoption increases by 8.2% if the farmers have access to information on climate change. This result is in line with Sardar et al. [77], who found that farmers are more likely to adopt CSAT if there is information on the destructive impacts of climate change. Interestingly, access to credit and extension services were found to influence the adoption of CSAT positively and significantly. The result suggests that a farmer that is well endowed with productive resources such as credit facilities, and has access to knowledge, skills, and awareness about the use of CSAT through extension services, is more likely to adopt than farmers who do not have access to such support. Studies [80–82] have confirmed that access to credit facilities and extension services are very important factors that mostly form farmers' opinions and decisions about adopting an agricultural technology.

The relationship between income from agriculture and the adoption of CSAT was found to be positive and significant at the 1% level. Income plays an important role in the decision-making process of most farming households [80]. The result suggests that an increase in the income generated from agriculture will lead to a 9.2% increase in the likelihood of adopting CSAT. Interestingly, distance to the nearest village was found to positively influence adoption at the 1% significant level. The marginal increase in the probability of adopting CSAT will be 1% in relation to the distance to the nearest village. As noted by Wang et al. [83], closeness and connections with agricultural hubs within farmers' localities could increase the likelihood of selling agricultural products, which may increase the probability of adoption of improved agricultural technology. Social capital is an important factor that influences individual farmers' decision to adopt an improved agricultural technology. We found that membership of any organization, such as a farmers' group, is positive and significantly influences the probability of adopting CSAT. The marginal effect estimates show that farmer membership of any social group increases

the likelihood of adopting CSAT by 11.9%. Studies [84–86] have found that the adoption rate of improved technology can be significantly increased if the household head belongs to any association.

### 3.4. The Distributional Effects of CSAT Adoption on Welfare Outcomes

Table 4 presents the results of the distributional impacts of the adoption of CSAT on the four welfare indicators considered in this study, which are the per capita total consumption expenditure, per capita non-food expenditure, per capita food expenditure, and per capita total household income. The results reveal that the treatment effects of the adoption of CSAT on per capita total consumption expenditure are positive and statistically significant at a 1% level across all the quantiles, except for the median (Q0.50). Specifically, the impacts of CSAT, in value terms, range between 11,399.70 CFA Franc for households at the lowest tail of the distribution to 46,902.43 CFA Franc for those at the highest tail. These findings reflect heterogeneity in the impacts of CSAT on welfare as measured by per capita total consumption expenditure.

**Table 4.** The distributional effects of CSAT adoption on welfare.

| Variable | IV-QTE Estimates | | | | |
|---|---|---|---|---|---|
| | Q0.15 | Q0.25 | Q0.50 | Q0.75 | Q0.85 |
| *Per capita total consumption expenditure (CFA)* | 11,399.70 *** | 13,981.77 *** | 29,217.44 | 41,897.82 *** | 46,902.43 *** |
| Treatment effect of CSAT adoption | (3395.17) | (3971.644) | (6794.67) | (10980.83) | |
| % Impact of CSAT adoption | 53.75 | 41.70 | 43.73 | 35.16 | 29.95 |
| *Per Capita non-food expenditure (CFA)* | 11,480.87 *** | 14,342.1 *** | 22,770.61 *** | 35,035.97 *** | 32,515.77 ** |
| Treatment effect of CSAT adoption | (3253.08) | (3724.21) | (6274.41) | (8557.16) | (15,257.77) |
| % Impact of CSAT adoption | 67.67 | 55.05 | 36.69 | 31.82 | 21.56 |
| *Per capita food expenditure (CFA)* | 367.45 ** | 491.15 *** | 686.23 *** | 1761.83 *** | 2530.44 *** |
| Treatment effect of CSAT adoption | (152.35) | (166.01) | (236.54) | (462.31) | (784.94) |
| % Impact of CSAT adoption | 31.20 | 31.89 | 24.77 | 34.57 | 39.37 |
| *Per Capita total household income (CFA)* | 5695.735 ** | 10,024.48 *** | 15,312.29 *** | 27,182.22 *** | 29,121.3 ** |
| Treatment effect of CSAT adoption | (2469.954) | (3358.808) | (4637.214) | (7047.017) | (12,355.48) |
| % Impact of CSAT adoption | 86.40 | 65.96 | 31.94 | 29.04 | 23.82 |

Note: The figures in parentheses are standard errors. *** $p < 0.01$, ** $p < 0.05$.

The adoption of CSAT significantly lead to an increase in the per capita total consumption expenditure by 53.75 and 41.70% for farming households in the 15th and the 25th quantiles, respectively, and 35.16 and 29.95% for farming households in the 75th and the 85th quantiles, respectively, implying that the impacts of CSAT on per capita total consumption expenditure are higher among poorer farm households compared to farm households that are well-off. This is in line with the finding of Olagunju et al. [50], who reported that the welfare outcomes of poorer maize farmers in rural Nigeria are more positively and significantly impacted by the adoption of improved seed varieties than those of well-off farmers.

The results also show that adoption of CSAT positively and significantly impacts both per capita food and non-food expenditure differently across the five quantiles, ranging from 11,480.87 (Q0.15) CFA Franc to 32,515.77 (Q0.85) CFA Franc for per capita non-food expenditure, and 367.45 (Q0.15) CFA Franc to 2530.44 (Q0.85) CFA Franc for per capita food expenditure. In terms of the percentage impact of CSAT, the findings show that the highest percentage increase in the effects of the adoption of CSAT was found at the lower tails of per capita food and non-food expenditure distributions. The adoption of CSAT significantly raised per capita non-food expenditure by 67.67 and 55.05% in the 15th and the 25th quantiles, respectively, and increased per capita food expenditure by 31.20 and 31.89% in the 15th and the 25th quantiles, respectively. The percentage impact of the adoption of CSAT on per capita non-food expenditure is significantly higher than the corresponding impact on per capita food expenditure. Given that the farm households' expenditure on non-food items is often larger than that on food items, the significantly larger impact of

CSAT on per capita non-food expenditure implies that adoption status will have a strong bearing on the livelihood status of rural farmers in the study area

Finally, the results show that the impact of the adoption of CSAT is also positive and significant across the distribution of the per capita total household income. In value terms, the IV-QTE estimates show a significant and increasing pattern along the per capita total household income distribution. The largest percentage impacts of about 86.40% in the Q.15 and 65.96% in Q0.25 were found in the lower quantiles of the per capita total household income distribution. This also suggests that the adoption of CSAT impacts per capita total income of poorer households substantially, in support of the existing findings on other welfare outcomes. Overall, all these findings are also similar to that [87] on the yield impact of Bt corn in the Philippines, which is strongly felt by farmers at the lower end of the yield distribution and points to the fact that the adoption of Bt corn, just like CSAT is 'pro-poor'.

## 4. Conclusions

This article investigates the distributional impact of the adoption of Climate-Smart Agricultural Technologies on four welfare indicators, namely, per capita total consumption expenditure, per capita non-food expenditure, per capita food expenditure, and per capita total household income. Adoption of CSAT is essential in achieving improvement in overall households' welfare. The analytical technique employed is based on the IV-QTE, which accounts for the selectivity bias attributed to unobservables.

The results from the analysis show that the farmers' decisions to adopt any CSAT are positively and statistically influenced by access to credit, contact with extension agents, participation in training, access to information through the television, and being a member of any organization, such as a cooperative society.

The results of the IV-QTE reveal heterogeneity in the impacts of CSAT on welfare, and the highest percentage increase in the impact of CSAT adoption was found at the lower tails of the per capita food expenditure consumption distribution. This implies that the impacts of CSAT on per capita food consumption expenditure are more pronounced among poorer farm households compared to farm households that are well-off. In the same vein, the highest percentage increase in the impact of CSAT adoption was found at the lower tails of per capita food and non-food expenditure distributions, and adoption of CSAT substantially impacts per capita total income of poorer households.

The findings offer important policy implications for agricultural policy development in Mali and other developing countries. The fact that the results show that the highest positive impacts of CSAT were found at the lower tail of the distribution of the welfare outcomes emphasizes that the development, dissemination, and adoption of CSAT are effective in addressing poor farming households, hence supporting the general knowledge that climate-smart technologies are pro-poor [88]. To encourage the adoption of CSAT, findings from this study suggest that policy efforts should promote awareness of CSAT through training and various extension outlets, particularly among the resource-poor farm households that are currently facing the adverse effects of climate change.

While the present study has made valuable additions to the literature, it is not without limitations. Firstly, it is important to note that the study was carried out only in the CSAT project-selected areas. Therefore, the findings from the cross-sectional data are limited to these localities, and cannot be generalized to the entire population of Mali. Future research may need to consider extending the research analysis beyond the selected areas. Additionally, the present study employs cross-sectional data and hence could not control for factors that are time-varying. As data become available, future research may consider looking into analyzing the distributional impact of CSAT using a longitudinal dataset.

**Author Contributions:** Conceptualization, B.A.A. and K.O.O.; methodology, B.A.A., A.O., K.O.O., L.O.B., A.Y.C., A.N.W., V.M. and T.A.; software, B.A.A., A.O., K.O.O., L.O.B., A.Y.C. and A.N.W.; validation, B.A.A., A.O., K.O.O., L.O.B., A.Y.C., A.N.W., B.K., A.A., V.M. and T.A.; formal analysis, B.A.A., A.O. and K.O.O.; investigation, B.A.A., A.O., K.O.O., L.O.B., A.Y.C., A.N.W., B.K., A.A, V.M. and T.A.; resources, V.M., B.A.A. and T.A.; data curation, B.A.A., A.Y.C. and T.A.; writing—original draft preparation, B.A.A., A.O. and K.O.O.; writing—review and editing, B.A.A., A.O., K.O.O., L.O.B., A.Y.C., A.N.W., V.M. and T.A.; visualization, B.A.A., A.O., K.O.O., L.O.B., A.Y.C., A.N.W., V.M. and T.A.; supervision, B.A.A., V.M. and T.A.; project administration, B.A.A. and T.A.; funding acquisition, V.M., B.A.A. and T.A. All authors have read and agreed to the published version of the manuscript.

**Funding:** This project (Climate-smart Agricultural Technologies for Improved Rural Livelihoods and Food Security (CSAT-Mali) in Mali), and the APC was funded by the Royal Norwegian Embassy in Mali (Grant MLI-17-0008).

**Institutional Review Board Statement:** Not applicable.

**Data Availability Statement:** The data for this study is available upon submitting a written request to the International Institute of Tropical Agriculture (IITA).

**Acknowledgments:** The authors acknowledged the financial support of the Royal Norwegian Embassy, Bamako, Mali. We appreciate the farmers in the CSAT project regions (Kayes, Koulikoro, Segou, and Sikasso) for their cooperation and willingness to share their ideas. We are also grateful to the collaborating national partners (ICRISAT, IER and NGOs-MaliMARK, AMEDD, and AMASSA Afrique vert).

**Conflicts of Interest:** The authors declare no conflict of interest.

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
