# Peer review of "Evaluating the Heterogeneous Impacts of Adoption of Climate-Smart Agricultural Technologies on Rural Households’ Welfare in Mali"

_agriculture, doi:10.3390/agriculture12111853_

Round 1

Reviewer 1 Report

I enjoyed by reading this article. It explains the Evaluating the Heterogeneous Impacts of Adoption of Climate-Smart Agricultural Technologies on Rural Households Welfare in Mali by using an instrumental variable quantile treatment effects model, the paper addresses the potential endogeneity arising from the selection bias and the heterogeneity of the effect across the quantiles of the outcome variables’ distribution.

Over manuscript is written well and have new insight i.e., improved households’ welfare and the farmers’ decision to adopt any CSAT, but need minor correction as following.

 Title: title need little modification and I suggest as following.

Evaluating the Heterogeneous Impacts of Climate Smart Agricultural Technologies Adoption on Rural Households’ Welfare in Mali

 Abstract:         I suggest, to write background of the study in one or two sentence(s).

                     Line 26~29, sentence, “The results show that adoption of CSAT ……. as cooperative society” is long, it is suggested, divide into tow sentences.

                    Introduction: The first is almost 5 years old, it is suggested to update the reference.

                     Line 74-~76, sentence, “Notably among these factors ……..climate events” need to revise grammatically.

                     Line 81~85, sentence “being a landlocked …….in general” is too long, need to divide into two or more sentences.

                     Line 92~95, sentence, “It is built upon ….. investments” is too long, need to divide into two sentences.

                     Line 128~133, sentence, “furthermore, if on the other hand, ….. farm household welfare” is too long, need to divide into two sentences. Further, authors stated in this sentence, “we discovered that adoption of CSAT significantly impacts the productivity, income, etc., and also suggestion has been given promoting the adoption of CSAT …”. I’m confused, either this sentence is part of the findings of this study or borrowed from other study. If this statement is part of this study, I suggest, delete from the introduction section and only keep it in the results and, or in conclusion sections.

Materials and Methods:          Why authors selected this region / area to study?

Results and Discussion:          Authors results stated that large farm size found negative association with adoption of CSAT, which is generally not happened. Large farm size household are willingness to adopt new technology. please interpret more about it.

                     Section 3.4. The distributional effects of CSAT adoption on welfare Outcomes: authors didn’t compare their results with existing literature, it would be better to give reference and compare with other recent studies.

Conclusion: Recommendation should extend regarding that how poor-resource farmers can access more financial support to enhance their crop yield, production and ultimately income for their welfare by adopting CSAT.

References: reference list is fine but need to cite latest in the section 3.4 to compare study results.

Author Response

Comments and Suggestions for Authors
-Reviewer 1

I enjoyed reading this article. It explains the Evaluating the Heterogeneous Impacts of Adoption of Climate-Smart Agricultural Technologies on Rural Households Welfare in Mali by using an instrumental variable quantile treatment effects model, the paper addresses the potential endogeneity arising from the selection bias and the heterogeneity of the effect across the quantiles of the outcome variables’ distribution.

 Over manuscript is written well and has new insight i.e., improved households’ welfare and the farmers’ decision to adopt any CSAT, but need minor correction as follows.

Response: We appreciate the reviewer for taking the time to read this manuscript and for the valuable comments/suggestions.

S/No.

Reviewer’s Comments

Authors Response

1

Title: title needs little modification and I suggest as follows: Evaluating the Heterogeneous Impacts of Climate Smart Agricultural Technologies Adoption on Rural Households’ Welfare in Mali.

The suggested modification in the title has been addressed accordingly. ….. Households’

2

 Abstract: I suggest, writing the background of the study in one or two sentences (s).

Thanks for this suggestion. We have added a very concise background to the abstract. Please see lines 23-25.

3

Line 26~29, sentence, “The results show that adoption of CSAT ……. as cooperative society” is long, it is suggested, divide into two sentences.

The sentence has been separated/divided into two sentences as suggested by the reviewer.

4

Introduction: The first is almost 5 years old, it is suggested to update the reference.

The first reference cited under the introduction has been updated to the year 2022. ‘The global population is rapidly growing and is projected to reach  8.0 billion people in 2022 [1]’. See lines 45-46.

5

Line 74-~76, sentence, “Notably among these factors ……..climate events” need to revise grammatically.

The sentence has been revised to correct the grammatical error. See lines 77-79

6

Line 81~85, sentence “being a landlocked …….in general” is too long, and needs to divide into two or more sentences.

This sentence has been restructured to reduce the length as suggested

7

Line 92~95, sentence, “It is built upon ….. investments” is too long, and needs to divide into two sentences.

A missing full stop has been inserted where necessary to cut the sentence into two.

8

Line 128~133, sentence, “furthermore, if on the other hand, ….. farm household welfare” is too long, needs to divide into two sentences.

The sentence has been separated into two sentences as suggested.

9

Further, the authors stated in this sentence, “we discovered that adoption of CSAT significantly impacts the productivity, income, etc., and also suggestion has been given promoting the adoption of CSAT …”. I’m confused, either this sentence is part of the findings of this study or borrowed from other studies. If this statement is part of this study, I suggest, delete from the introduction section and only keep it in the results and, or in the conclusion sections.

This is part of the justification for the study and was inserted to explain the potential/expected results from the study and the implications.

10

Materials and Methods: Why authors selected this region/area to study?

The regions, communes, and villages for the CSAT project were purposively selected based on many criteria. We looked at the geographical location of the region in terms of exposure to negative effects of climate change, the security, proximity to the main road, accessibility, and willingness of the people to participate in the project; mainly the readiness of the households to release a portion of their land to be used for demonstrations.

11

Results and Discussion: Authors’ results stated that large farm sizes found a negative association with the adoption of CSAT, which is generally not happened. Large farm-size households are willing to adopt new technology. please interpret more about it.

In some cases, depending on the locality large farm size can have a negative association with technology adoption. The experience from the field reveals that in most cases, the farmers with large farm sizes don’t adopt new improved technologies on the entire farmland. Farmers are risk averse, and hence new technologies are planted/adopted on a smaller portion of their land.

12

Section 3.4. The distributional effects of CSAT adoption on welfare Outcomes: authors didn’t compare their results with existing literature, it would be better to give reference and compare them with other recent studies.

Thank you for the important suggestion. The missing comparison has been added: This finding is also similar to that of Sanglestsawai et.al (2014) on the yield impact of Bt corn in the Philippines which is strongly felt by farmers at the lower end of the yield distribution and like the adoption of CSAT the result shows that the adoption of Bt corn is ‘pro-poor’. see lines 511-514.

13

Conclusion: Recommendation should extend regarding that how poor-resource farmers can access more financial support to enhance their crop yield, production, and ultimately income for their welfare by adopting CSAT.

We have based the recommendations on the main findings from the study. We acknowledge the fact that financial support is critical for smallholders’ productivity increase, but this study did not include this aspect in our analyses.

References: reference list is fine but needs to cite latest in the section 3.4 to compare study results.

We have added some latest references as suggested.  On the overall, all these findings are also similar to that of Sanglestsawai et.al (2014) on the yield impact of Bt corn in the Phillipines which  is strongly felt by farmers at the lower end of the yield distribution and pointed to the fact  that the adoption of Bt corn, just like CSAT is ‘pro-poor’. see lines 511-514.

Reviewer 2 Report

This is a fascinating piece of work that would contribute to the expanding body of knowledge regarding CSAT.

I suggest the authors add a few details to the Introduction Section about what has been discovered and discussed in relation to the CSAT research that has been conducted thus far in Mali. This would make it crystal obvious which gaps this research fills.

Author Response

  Comments and Suggestions for Authors
-Reviewer 2

Comment: This is a fascinating piece of work that would contribute to the expanding body of knowledge regarding CSAT.

Response: We are very grateful for the reviewers’ time and nice comments.

Response:

S/No.

Reviewer’s Comment

Author Response

1

I suggest the authors add a few details to the Introduction Section about what has been discovered and discussed in relation to the CSAT research that has been conducted thus far in Mali. This would make it crystal obvious which gaps this research fills.

Thank you so much for this important suggestion. We have added the most relevant literature on climate-smart technologies from Mali. ( Ouédraogo, et.al. (2019). Uptake of Climate-Smart Agricultural Technologies and Practices: Actual and Potential Adoption Rates in the Climate-Smart Village Site of Mali). See lines 105-113.

Ouédraogo, et.al. (2019). Uptake of Climate-Smart Agricultural Technologies and Practices: Actual and Potential Adoption Rates in the Climate-Smart Village Site of Mali. sustainability 2019, 11(17), 4710; https://doi.org/10.3390/su11174710

Reviewer 3 Report

Dear authors,

thanks for the opportunity to read your research, which I find very appealing. The topic of the article is up to date.These observations will help to guide your efforts to improve the quality of the paper:

1. Rows 104, 115, 211 - please revise and correct.

2. The Introduction section is well developed but I think that a Literature Review section is also important.

3. Missing hypotheses that must be grouded on concrete literature.

4. Missing limitations of the research.

5. The research implications should be clearly provided for both research and practice.

Author Response

Comments and Suggestions for Authors
-Reviewer 3.

Dear authors,

Thanks for the opportunity to read your research, which I find very appealing. The topic of the article is up to date. These observations will help to guide your efforts to improve the quality of the paper:

Response: Thank you so much for your time and encouraging comment. We are sincerely grateful.

Table 1: Responses to Reviewer’s Comments

S/No

Reviewer’s Comments

Authors’ Response

1

Comment1: Rows 104, 115, 211 - please revise and correct

All the errors identified have been corrected. All the necessary corrections are put in track changes see pages 5 and 6; lines

2

The Introduction section is well developed but I think that a Literature Review section is also important.

To make the paper short and concise, we have submerged the necessary literature inside the introduction. We think that creating a section on the literature review will not be needed as it will make the manuscript unnecessarily long.

3

Missing hypotheses that must be grounded on concrete literature

As suggested, we have included the study hypothesis and placed it within the context of study motivation and previous studies. See Lines 157 – 161.

“ Specifically, we hypothesize that the impacts of CSAT adoption differ across the distributions of our welfare indicators including income, per capita total expenditure, and food and non-food expenditure.”

4

Missing limitations of the research

We agree with this comment and have included the limitations of our research and suggestions for future studies. See the last paragraph of the conclusion section – Lines 535 – 542.

“While the present study has presented valuable additions to the literature, it is not without limitations. Firstly, it is important to note that the study was carried out only in the CSAT project-selected areas. Therefore, the findings are limited to these localities, and cannot be generalized to the entire population of Mali. Future research may consider extending the research analysis beyond the selected areas. Besides, the present study employs cross-sectional data and hence could not control for factors that are time-varying. As data becomes available, future research may look into analysing the distributional impact of CSAT using longitudinal dataset.”

5

The research implications should be clearly provided for both research and practice.

conclusions are insufficient, and the whole section needs development. 

We have substantially revised the conclusion section to clearly state the implication of our findings for policy considerations. Please, see lines 522- 557.

Reviewer 4 Report

The reviewed paper is of interest for the journal and potential publication but is still in a premature stadium. I recommend the following major corrections:

- the bibliography relies too heavily on local publications and grey literature. The authors need to complement their references with peer-reviewed international studies on climate change, environmental and resource innovation, agricultural poverty, vulnerability, resilience and sustainability and rural development. Check for publications from Rusciano, Aldieri, Gatto and colleagues.

- addressing the previous issue will help to better frame the paper within the existing literature which is currently deficient.

- add the paper's organisation at the end of section 1.

- citations need to be discussed - see, for instance "Concerning CSAT practices, there exists extensive 102 literature on the adoption impacts of individual climate-smart practices, although with 103 divergent findings [e.g., 30-40].". The authors cannot simply mention previous studies.

- the paper does not really discuss the sorting results. A discussion section/sub-section needs to be implemented.

- conclusions are insufficient, the whole section needs development. 

- check for grammar errors, inconsistencies and existing typos.

Author Response

Comments and Suggestions for Authors
-Reviewer 4

The reviewed paper is of interest to the journal and potential publication but is still in a premature stadium. I recommend the following major corrections:

Response: We appreciate the reviewer for the comments.

S/No.

Reviewer’s Comment

Author’s Response

1

The bibliography relies too heavily on local publications and grey literature. The authors need to complement their references with peer-reviewed international studies on climate change, environmental and resource innovation, agricultural poverty, vulnerability, resilience and sustainability, and rural development. Check for publications from Rusciano, Aldieri, Gatto, and colleagues.

- addressing the previous issue will help to better frame the paper within the existing literature which is currently deficient.

We appreciate the reviewer for the invaluable comments and the suggested literature. However, we could not locate the suggested literature, but we were able to add other very relevant ones. Please, see lines 60-61

1.       Paris and Rola-Rubzen, 2018;

2.       Anuga et. Al. 2019), and

3.       Bitterman et.al. 2019

2

Add the paper's organization at the end of section 1.

Thank you for the comment. The organization of the paper has been included at the end of section 1. See lines 189-191.

3

Citations need to be discussed - see, for instance, "Concerning CSAT practices, there exists an extensive literature on the adoption impacts of individual climate-smart practices, although with divergent findings [e.g., 30-40].". The authors cannot simply mention previous studies.

The missing literature and the suggested discussions have been added. Please,  see lines 114-122.

4

The paper does not really discuss the sorting results. A discussion section/sub-section needs to be implemented.

We think it is better to present the result and discussion together instead of having two separate sections. This allows readers to have a quick and easy understanding of the results and the interpretation at the same time. Hence, we have a section on results and discussion.

5

Conclusions are insufficient, and the whole section needs development. 

The conclusion section has been improved as suggested. Please, see lines 507-545

6

Check for grammar errors, inconsistencies, and existing typos.

Thank you so much. This has been checked, throughout the manuscript.

Round 2

Reviewer 1 Report

Author(s) have improved the revised version of manuscript. I'm satisfied with the response to the comments.